# Smoking Suppresses the Therapeutic Potential of Adipose Stem Cells in Crohn’s Disease Patients through Epigenetic Changes

**DOI:** 10.3390/cells12071021

**Published:** 2023-03-27

**Authors:** Albert Boronat-Toscano, Irene Vañó, Diandra Monfort-Ferré, Margarita Menacho, Gemma Valldosera, Aleidis Caro, Beatriz Espina, Maria José Mañas, Marc Marti, Eloy Espin, Alfonso Saera-Vila, Carolina Serena

**Affiliations:** 1Hospital Universitari de Tarragona Joan XXIII, Institut d’Investigació Sanitària Pere Virgili, Universitat Rovira i Virgili, 43007 Tarragona, Spain; 2Digestive Unit, Hospital Universitari Joan XXIII, 43007 Tarragona, Spain; 3Colorectal Surgery Unit, Hospital Universitari Joan XXIII, 43007 Tarragona, Spain; 4Colorectal Surgery Unit, Hospital San Pau i Santa Tecla, 43003 Tarragona, Spain; 5Colorectal Surgery Unit, General Surgery Service, Hospital Vall de Hebron, Universitat Autonoma de Barcelona, 08035 Barcelona, Spain; 6Sequentia Biotech, Carrer Comte D’Urgell 240, 08036 Barcelona, Spain

**Keywords:** DNA methylation, tobacco, cigarette, macrophages, cell therapy, adipose tissue, inflammatory bowel diseases, chronic inflammatory disease, immune cells, differentially methylated regions

## Abstract

Patients with Crohn’s disease (CD) who smoke are known to have a worse prognosis than never-smokers and a higher risk for post-surgical recurrence, whereas patients who quit smoking after surgery have significantly lower post-operative recurrence. The hypothesis was that smoking induces epigenetic changes that impair the capacity of adipose stem cells (ASCs) to suppress the immune system. It was also questioned whether this impairment remains in ex-smokers with CD. ASCs were isolated from non-smokers, smokers and ex-smokers with CD and their interactions with immune cells were studied. The ASCs from both smokers and ex-smokers promoted macrophage polarization to an M1 pro-inflammatory phenotype, were not able to inhibit T- and B-cell proliferation in vitro and enhanced the gene and protein expression of inflammatory markers including interleukin-1b. Genome-wide epigenetic analysis using two different bioinformatic approaches revealed significant changes in the methylation patterns of genes that are critical for wound healing, immune and metabolic response and p53-mediated DNA damage response in ASCs from smokers and ex-smokers with CD. In conclusion, cigarette smoking induces a pro-inflammatory epigenetic signature in ASCs that likely compromises their therapeutic potential.

## 1. Introduction

Crohn’s disease (CD) is a heterogeneous disorder characterized by chronic inflammation and ulceration of the small or large bowel and is associated with poor quality of life. Its etiology is not fully understood, but it is thought that both genetic and environmental factors contribute to the physiopathology of the disease [1]. Currently, there are CD patients unresponsive to available treatments such as immunomodulators, corticoids and biological treatments [2]. Despite improvements in the medical management of CD, more than two-thirds of patients require intestinal resection. In most of these cases, patients ultimately receive ileocecal resection surgery due to inflammatory lesions, which frequently recur on the anastomosis and/or on the neo-terminal ileum [3]. Moreover, post-operative recurrence is frequently observed. Environmental factors can also increase the risk of post-operative recurrence, and exposure to cigarette smoke is considered as one of the main risk factors for post-operative recurrence as well as for the progression of the disease per se [4].

In recent years, stem cell transplantation has been shown to be useful for refractory luminal CD, reducing gut inflammation [5]. In addition, allogenic mesenchymal adipose-derived stem cells (ASCs) isolated from the subcutaneous adipose tissue (SAT) have demonstrated promising results in the treatment of complex perianal fistulae, a common complication of CD that rarely heal without treatment [6]. ASCs have important roles in tissue development and in the maintenance of cell homeostasis, tissue repair and regeneration. ASCs can also attenuate immune system response through their immunomodulatory activity, which is thought to be dependent on two different mechanisms: direct cell–cell interactions and secretion of soluble factors. The first mechanism involves the direct communication between ASCs and immune cells, which drives anti-inflammatory polarization (M2) of activated monocytes and inhibits the proliferation of T- and B-lymphocytes [7]. The second mechanism involves the secretion of soluble mediators such as anti-inflammatory cytokines, which modulate immune responses through paracrine mechanisms.

It has been previously reported that the immune properties of ASCs are compromised in patients with CD [8]. Specifically, ASCs isolated from the mesenteric SAT of patients with CD display an enhanced proliferative, inflammatory, invasive and phagocytic phenotype when compared with equivalent cells from healthy donors. Additionally, these cells show an activated immune response that was associated with a reduction in their immunosuppressive properties. Beyond the disease per se, ASCs properties can be affected by other factors including the body mass index (BMI) [9] or aging [10]. Moreover, lifestyle habits can also disturb the metabolic and cell growth-signaling pathways of ASCs, which affects their phenotype and function [11]. For instance, obesity is associated with mesenchymal stromal cell senescence [12] whereas physical exercise is associated with increased all-round “potency” [13].

It is widely known that cigarette smoke exposure increases the risk of various cancers and systemic diseases [14,15]. Indeed, recent studies claim that persistent smoking causes lasting effects on the DNA methylation pattern [16], one of the well-explored epigenetic modifications that DNA can undergo. The enzyme DNA methyltransferases (DNMTs) are responsible for these alterations, and as a result of DNMTs, the 5-methylcytosine existing in CpG islands can be methylated into DNMTs [17]. This epigenetic modification might lead to changes in gene expression and, in a broader context, to the development or progression of various diseases including systemic diseases and cancer. Smokers with CD are significantly more likely to experience disease recurrence and typically have more severe symptoms and complications [18]. For instance, they are more likely to develop fistulas, require more immunosuppressant drugs and are more likely to need surgery. Accordingly, encouraging patients to quit smoking is an important part of the management of CD.

Given this background, it was hypothesized that smoking affects the immune therapeutic potential of ASCs from CD patients. To address this issue, we first studied if the immune regulatory properties of ASCs isolated from the SAT of these patients were affected by smoking. Secondly, we questioned whether these alterations can be explained by epigenetic changes in ASCs because of the smoking habits and if these changes were sustained over time. It would be important to understand how factors such as the smoking habit can affect the biological efficacy of ASCs, even as an exclusion criterion when designing future stem cell-based trials and therapies [18].

## 2. Materials and Methods

### 2.1. Study Subjects

Subjects were recruited at the University Hospital Joan XXIII (Tarragona, Spain) and University Hospital Vall d’Hebrón (Barcelona, Spain). The study was conducted according to the tenets of the Helsinki Declaration and was approved by the ethics committees of each hospital (references CEIM 177/2018 and PR[CS]383/2021, respectively). All participants signed an informed consent. Patients were classified as those in relapse (active) or remission (inactive) following criteria of the Crohn’s Disease Activity Index (CDAI) and also based on clinical and biological parameters such as high-sensitivity C-reactive protein (hsCRP). Endoscopic evaluation was performed in 75% of the patients, with complete correspondence with the clinical classification obtained by CDAI [19,20]. Inactive CD patients and non-IBD (healthy controls) were recruited from subjects undergoing non-acute surgical interventions such as hernia or cholecystectomy, scheduled as a routinary surgery, whereas patients with active CD were recruited from those undergoing surgery for symptomatic complications.

Samples of SAT were aseptically collected from patients classified as non-smokers, smokers or ex-smokers depending on their smoking habit. Clinical data, anthropometric, demographic and biochemical variables of the cohort are shown in Appendix A. No significant differences were found among the groups.

### 2.2. Adipose Tissue-Derived Stem Cells

Human SAT biopsies were first washed with PBS to remove any traces of blood, and when needed, the parts damaged by the extraction process were removed with a scalpel. The tissue was then treated with 0.1% collagenase/1% BSA in PBS for 90 min at 37 °C under gentle agitation. The digested samples were centrifuged at 300× *g* at 24 °C for 5 min to obtain phase separation, with mature adipocytes floating to the top and the vascular stroma in the pellet, consisting mainly of stem cells, macrophages, endothelial cells and blood cells. Another wash and centrifugation cycle were performed to isolate ASCs. The pellet resulting from the centrifugation was resuspended with DG maintenance medium (comprising DMEM/F12 in 10% fetal bovine serum [FBS]) in a T75 flask and allowed to grow for seven days in a 37 °C incubator to 90% confluence. Cells were then harvested with trypsin-EDTA, and were considered as passage 0 (P0) [21]. DNA for methylation assays and RNA for gene expression studies were extracted from cells at P3.

### 2.3. Adipose Stem Cell Immunophenotyping

Cells (2 × 10^5^) were incubated with a panel of primary antibodies (described in Appendix A) [22]. Cells were incubated for 20 min at RT (protected from light) with the following antibodies: fluorescein isothiocyanate (FITC)-conjugated antibodies to human cluster of differentiation (CD) 34; PeCy5-conjugated antibodies to human CD105 and CD90; PeCy7-conjugated antibodies to human CD45 and CD73. Finally, ASCs were suspended in 400 mL of Hank’s balanced salt solution without calcium and magnesium, and the vital dye Hoechst 33342 (Ho342; 40 mg/mL final concentration) was added to discriminate nucleated cells. Samples were kept in the dark at 4 °C until analysis. Data were acquired on FACSAria™ III (BD Biosciences, Franklin Lakes, NJ, USA) and were analyzed using FACSDiva™ v8 (BD Biosciences, Franklin Lakes, NJ, USA) and FlowJo™ v10 software (Ashland, OR, USA). At least 20,000 events were analyzed in each run.

### 2.4. THP-1, JURKAT and MEC-1 Cell Culture

The human monocyte cell line THP-1 and the human T-lymphocyte cell line Jurkat were obtained from ATCC (Rockville, MD, USA). The chronic lymphocytic leukemia cell line MEC-1 was obtained from Deutsche Sammlung von Mikroorganismen und Zellkulturen (DSMZ), GmbH (Braunschweig, Germany). All cells were maintained as described [8]. Briefly, cells were cultured in RPMI-1640 medium supplemented with 10% FBS and 1% antibiotic/antimycotic solution in a humidified incubator at 37 °C with 5% CO_2_. The medium was half-replenished every 2–3 days.

### 2.5. Stimulation of THP-1 Cells

THP-1 monocytes were differentiated to macrophages with 10 ng/mL phorbol-12-myristate-13-acetate (PMA) for 72 h in 6-well culture plates with 0.5 mL of cell suspension (5 × 10^5^ cells) in each well. Differentiated cells were washed and then incubated for 48 h in the culture medium to return to the resting macrophage state (M0). Subsequently, 24 h-conditioned medium (CM) from ASCs were added in duplicate in each well and gene expression studies were performed 24 h later.

### 2.6. RNA Extraction

RNA was extracted from ASCs (100,000 cells/well) or THP-1 (90,000 cells/well) using the TriPure Isolation Reagent (Roche, Basel, Switzerland). RNA concentration was determined by absorbance at 260 nm, and purity was estimated with a Nanodrop spectrophotometer (Nanodrop Technologies Inc., Wilmington, DE, USA). The purity of each extraction was determined by the OD260/OD280 ratio. cDNA was synthesized using SuperScript II reverse transcriptase and random hexamer primers (Invitrogen Life Technologies, Darmstadt, Germany).

### 2.7. Real-Time Quantitative PCR

Quantitative gene expression was evaluated by quantitative real-time PCR (qPCR) on a 7900HT Fast Real-Time PCR System using the TaqMan Gene Expression Assay (Applied Biosystems, Foster City, CA, USA). Results were calculated using the comparative threshold cycle (Ct) method (2−ΔΔCt) normalized to the expression of the housekeeping gene 18S (Hs03928985_g1) and expressed relative to the control condition, which was set to 1. Two technical duplicates were performed for each biological replicate. The TaqMan probes used are listed in Appendix A.

### 2.8. T- and B-Cell Proliferation Assay

Proliferation of Jurkat and MEC-1 cells in response to 24 h-CM from ASCs was assessed by BrdU incorporation using the BrdU Cell Proliferation Assay Kit (Millipore, Billerica, MA, USA) as described [8].

### 2.9. Analysis of Differentially Methylated Loci

A subcohort of our previous gene methylation profiling data sets deposited in the GEO database (accession code GSE138311) was used in which smoking habit information was included for each sample [23]. Briefly, global DNA methylation profiles determined with the Human MethylationEPIC BeadChip Infinium assay were used (Illumina Inc., San Diego, CA, USA), developed to quantitatively assay over 850,000 methylation sites across the genome at single-nucleotide resolution. All pre-processing and statistical data treatment were performed with the R statistical programming environment (version 4.2.0; R core team, 2022). DNA methylation raw data (.idat files) were read using the Bioconductor package RnBeads (v2.14) and the annotation package RnBeads.hg19 (v1.28) [24,25,26]. A total of 18.56% of the probes were discarded as they were identified as SNP-enriched probes; containing low detection *p*-values or standard deviation; were outside the GpG context or within sex chromosomes; or were included in a blacklist as putative error-prone probes (Product Change Notification, PCN2019-019). The beta-mixture quantile normalization (bmiq) algorithm was used to normalize probe intensities [27]. CpG methylation levels were calculated as M-and β-values. Differentially methylated positions (DMPs) were located through linear regression using the dmpFinder function from the Minfi R package and were calculated using default settings. As the sample number was lower than 10, variance shrinkage was set to TRUE to squeeze sample variances by computing empirical Bayes posterior means with the limma package, as recommended [28]. Based on Illumina’s manifest, DMPs were assigned to genes and candidate genes were identified based on a nominal *p* value cutoff < 1 × 10^−^^4^. Differentially methylated regions (DMRs) were identified using the bumphunter function from Minfi library package [29] using an inclusion threshold to 0.2. To remove non-biologically significant DMRs, they were filtered for at least three consecutive CpGs in 1 Kb within a DMR. Once more, significant DMRs were allocated to genes based on Illumina’s manifest, and candidate genes were chosen based on the amount of DMRs per gene and their relationship to the gene structure, as well as the *p* value cutoff of 0.05.

### 2.10. Functional Network Analysis and Visualization

Gene Ontology (GO) enrichment analysis and visualization were performed using the clusterProfiler R package [30]. Multiple test correction was performed with the false discovery rate (FDR) method [31] and the significance level was set at FDR < 0.05. Networks of the biological processes implicated in the GO enrichment analysis were performed using STRING v.11.00 software: protein–protein association network accessed on 30 January 2023 (https://string.db.org/).

### 2.11. Statistical Analysis

Statistical analyses were performed at different levels using multiple tools. For methylome analysis, R statistical programming language with the aforementioned packages as well as ggplot2 [32] were applied for visualization of the results. For clinical and anthropometrical variables, normally distributed data were expressed as mean ± SD; for variables with no Gaussian distribution, values were expressed as median (25th–75th quartiles). Student’s *t*-test with Bonferroni adjustment was used to compare the mean value of normally distributed continuous variables. For variables that did not have a Gaussian distribution, the Kruskal–-Wallis test with *post hoc* Dunn’s multiple comparisons test was used. To analyze the differences in nominal variables between groups, the χ2 test was performed. Pearson’s correlation coefficient with Bonferroni adjustment was used to analyze the relationship between parameters. The Statistical Package for the Social Sciences software, version 15 (SPSS, Chicago, IL, USA) for analysis was utilized. For in vitro data, experimental results are presented as mean ± SEM. Comparisons among the three groups were performed using two-way ANOVA followed with the Kruskal–Wallis test. Visualizations were performed with GraphPad Prism v6 (Graphpad Software Inc., San Diego, CA, USA).

## 3. Results

### 3.1. Smoking Disrupts the Immune Regulatory Properties of Adipose-Derived Stem Cells from Patients with Crohn’s Disease

It was examined whether smoking status impacted the immune regulatory properties of ASCs isolated from the SAT of patients with CD. Firstly, it was studied the relative gene expression of pro-inflammatory markers (*TNFA*, *IL1B* and *CASP1*) and inflammatory cytokine secretion (IL1B) in ASCs of the three groups (Figure 1A, left panel). No significant differences were observed in *TNFA* gene expression between the three groups, but *IL1B* gene expression was significantly higher in ASCs from the smoker and ex-smoker groups than from the non-smoker group. This was confirmed by examining IL1B secretion levels in the CM from ASCs, which were also significantly greater in the smoker and ex-smoker groups. *CASP1* gene expression in the three groups was like *IL1B* expression, but significant differences were only observed between ASCs from ex-smokers and non-smokers (Figure 1A, right panel). In addition, expression levels of anti-inflammatory gene *TGFB1* (Figure 1B) were measured. Notably, *TGFB1* expression was considerably lower in the smoker and ex-smoker groups than in the non-smoking group.

Next, it was questioned whether smoking status impacted the ability of ASCs to polarize M0 macrophages to the M2 anti-inflammatory phenotype. To do this, the relative gene expression of inflammatory M1 macrophage (*TNFA*, *IL6* and *IL1B*) and M2 macrophage (*IL10*, *CD163* and *MRC1*) markers were assessed in PMA-stimulated M0 macrophages cultured with 24 h-CM from ASCs of the three groups (Figure 1C). As expected, the CM of ASCs from the non-smoker group significantly increased the expression of M2 markers (*IL10*, *CD163* and *MRC1*) and decreased the expression of the pro-inflammatory cytokine *IL1B* in macrophages. A similar trend was found for *TNFA* and *IL6*. These findings indicate that ASCs from non-smoker patients can polarize M0 macrophages to anti-inflammatory M2 macrophages. The CM of ASCs from the smoker group had the opposite effect, decreasing markedly the gene expression of the M2 macrophage markers *IL10*, *CD163* and *MRC1* and significantly increasing the expression of the M1 markers *TNFA*, *IL6* and *IL1B.* Notably, the CM of ASCs from the ex-smoker group behaved similarly to that of the smoker group, decreasing the expression of all M2 anti-inflammatory markers (*IL10*, *CD163* and *MRC1*) and increasing the expression of *IL1B*. These findings suggest that smoking behavior (both current and former smoking) disrupts the ability of ASCs to promote the M2 macrophage phenotype.

Finally, the ability of ASCs to inhibit B- and T-lymphocyte proliferation was tested (Figure 1D). Results revealed that, despite not reaching the inhibition levels of non-IBD ASCs (healthy controls), the non-smoker CD patients’ ASCs were the only ones, from all the CD-isolated ASCs, that were able to significantly inhibit Jurkat (T-cell) and MEC-1 (B-cell) proliferation. Thus, smokers and ex-smokers CD patients’ ASCs were unable to inhibit T- and B-cell proliferation and intriguingly prompt T-cell proliferation.

Overall, the findings strongly suggest that smoking dramatically disturbs the ability of ASCs from patients with CD to modulate the immune system response. Specifically, smoking habit induces the production and secretion of inflammatory cytokines, polarizes macrophages to the M1 pro-inflammatory phenotype and suppresses ASCs ability to inhibit the proliferation of T- and B-immune cells.

### 3.2. Smoking Induces Changes in the Methylation Pattern of Adipose-Derived Stem Cells from Patients with Crohn’s Disease

To study whether the evident changes to the immunosuppressive properties of ASCs were triggered by smoking-related epigenetic effects, it was analyzed the genome-wide DNA methylation profiles of a cohort of non-smoking healthy individuals (n = 3) and of patients with CD (n = 7) including non-smokers (n = 3), smokers (n = 2) and ex-smokers (n = 2). Clinical data, anthropometric, demographic and biochemical variables of the cohorts are shown in Appendix A.

Correlation analysis of the heat map revealed that smoking habit was a crucial factor to explain the epigenetic variations detected in ASCs (Figure 2A). Notably, a significant correlation with CD (having or not having the disease) and with BMI was also found, as previously reported [8]. Furthermore, principal component analysis (PCA) (Figure 2B and Appendix A) of the first two principal components (PC1 and PC2) of the methylation data revealed differences between ASCs of smokers, non-smokers and ex-smokers with PC1 explaining 19.91% of the variance and PC2 explaining 19.56%.

### 3.3. Smoking-Associated Differentially Methylated Positions in Adipose-Derived Stem Cells from Patients with Crohn’s Disease Related to Regenerative Properties

The DNA methylation data was first examined to identify DMPs and the analysis revealed 1776 significant DMPs among non-smoker, smoker and ex-smoker groups. The top 20 genes with the greatest DNA methylation changes (hyper- and hypomethylated) are shown in Appendix A. Gene Ontology (GO) enrichment analysis for the whole set of genes with significant DMPs showed important biological processes affected by epigenetic modifications (Figure 2C). Of note, two of the most significantly altered biological processes were related to the immunosuppressive properties of ASCs: wound healing (GO:0042060) with 32 genes involved, osteoblast differentiation (GO:0001649) with 16 genes involved, cell junction assembly (GO:0034329) with 30 genes, cell–substrate adhesion (GO:0031589) with 28 genes and extracellular matrix organization (GO:0030198) with 26 genes. (Figure 2D and Appendix A). To visualize the network of these two processes, a functional analysis using STRING v.11.00 was performed (Figure 2D). The list of all the enriched biological processes is in Appendix A.

### 3.4. Smoking Affects the Regenerative and Anti-Inflammatory Properties of ASCs, Which Endure after Smoking Cessation

The DMRs between two comparative groups were examined: smokers *versus* non-smokers and ex-smokers versus non-smokers. In the first comparison, 2596 DMRs were discovered, and in the second comparison, 4077 DMRs were discovered. In both cases >80% of the major alterations were observed in the promotor regions (Figure 3A). Then, a list of genes with DMRs that were common in both comparative groups (Figure 3B) was compiled. Of note, TNXB (Tenascin XB) had the highest number of DMRs with eight and twelve significant DMRs (smoker versus non-smoker and ex-smoker versus non-smoker, respectively). Other genes that shared DMRs in both smoking and ex-smoking patient ASCs compared with non-smoking patient ASCs included SDK1 (sidekick cell adhesion molecule 1), GSX2 (GS homeobox 2), EN1 (engrailed homeobox 1), PAX9 (paired box 9), ATP11A (ATPase Phospholipid Transporting 11A), EBF3 (early B-cell factor), HOXA3 (homeobox A3) and LMO3 (LIM Domain Only 3). Figure 3C summarizes the DMRs found in the gene TNXB in the comparison smoker versus non-smoker (graphic above) and ex-smoker versus non-smoker (graphic below). Notably, the promoter region of TNXB gene was hypermethylated in both comparisons. In addition, a GO enrichment analysis of genes with significant DMRs was performed between groups to explore the biological processes implicated between the two comparisons: smoker versus non-smoker (Figure 4A) and ex-smoker versus non-smoker (Figure 4B). Notably, the results of this analysis indicated that the smoker and ex-smoker groups shared a disruption in several processes caused by smoking including T-cell differentiation, response to transforming growth factor beta or DNA damage response, signal transduction by p53 class mediator. Interestingly, several biological processes related to wound healing such as cell junction assembly, cell–substrate adhesion, extracellular matrix assembly and extracellular matrix organization were also detected in the GO analysis. The list of all the GO-enriched terms are shown in Appendix A.

## 4. Discussion

It is known that patients with CD who smoke have a worse prognosis and are at higher risk of post-surgical recurrence than their non-smoking peers [33]. Moreover, patients who quit smoking after surgical intervention have a dramatically reduced risk of post-operative recurrence. It was found that smoking impaired the immunomodulatory properties of isolated ASCs, conditioning them to become pro-inflammatory, which would also have a detrimental impact on the symptomatology of the disease itself. There is consensus that a pro-inflammatory phenotype negatively affects proliferation, chondrogenic and osteogenic differentiation, and migratory properties of ASCs [11,34]. Indeed, it has been previously reported that smoking negatively impacts the immune properties in adipose stromal cells [35], although the effects of cigarette smoking on the immunoregulatory properties of ASCs from patients with CD were unknown.

This study demonstrated that cigarette smoke impairs the autocrine and paracrine immunomodulatory mechanisms through which ASCs dampen the immune system response, including the augmented expression/synthesis of pro-inflammatory mediators. Under physiological conditions, ASCs can inhibit the proliferation of B- and T-lymphocytes and can also polarize macrophages to an anti-inflammatory M2 phenotype [36,37]. However, ASCs from smokers and ex-smokers have lost this capacity and, in fact, enhance lymphocyte proliferation and polarize macrophages to a pro-inflammatory M1 phenotype. In most cases, ASCs from ex-smokers showed an intermediate phenotype between smokers and non-smokers, suggesting that quitting smoking might reduce the inflammatory phenotype, although anti-inflammatory markers were not restored.

To study whether these evident alterations in immunomodulatory properties were caused by the smoking habits of patients, an epigenetic analysis was performed. PCA revealed that the DNA methylation variability in ASCs could be explained, at least partly, by the smoking habit. This is similar to our previous studies where it was found that epigenetic changes in ASCs could be explained by BMI and by the disease [8,38]. Intriguingly, genes with different methylation patterns in smoker patients (present and past) were involved in relevant biological processes such as wound healing, osteoblast differentiation, T-cell differentiation, extracellular matrix organization and response to transforming growth factor beta, which are all crucial to ameliorate inflammation and regenerate damaged tissue. Interestingly, TGFB is only expressed in mammals and mainly secreted and stored as a latent complex in the extracellular matrix [39]. More importantly, its activation is necessary to recruit stem cells in order to participate in the tissue regeneration and remodeling process [40]. In our study, the gene expression of TGFB1 was significantly decreased in ASCs isolated from smokers and ex-smokers with CD. Notably, both groups exhibit differentially methylated genes compared with the non-smokers’ group involved in biological processes such as extracellular matrix organization, T-cell differentiation, and response to transforming growth factor beta. These links point to TGFB1 as a crucial player in the wound healing process mediated by ASCs which may be deregulated by cigarette smoking habit in patients with CD. Moreover, ASCs from smokers (but also ex-smokers) had impaired biological regulation of DNA damage response, signal transduction by p53 class mediator, which in fact is likely important for genomic integrity and protection against smoke-induced DNA damage.

Changes in methylation patterns can have far-reaching consequences for health [41,42]. Cigarette smoking altered the methylome of ASCs from patients with CD. In a first approach, DMPs analysis between non-smoker, smoker and ex-smoker groups revealed significant differences in ETS1 and RUNX2, which are involved in wound healing and osteoblast differentiation, respectively. Interestingly, ETS1 plays a critical role in regulating the TGF-β pathway in mesenchymal cells and its overexpression is related to aggressive behavior in tumors [43]. Furthermore, cigarette smoke is known to alter the expression of ETS1 [44]. ETS1 has also been linked to CD, as it is highly expressed in CD4+ T-cells from patients with inflammatory bowel disease and promotes Th1-driven mucosal inflammation through cold-inducible RNA binding protein (CIRBP) [45]. These changes would likely contribute to worse CD outcomes. In accord with our analysis, a previous study showed that exposure of healthy ASCs to cigarette smoke extract upregulated the expression of RUNX2, which reduced the capacity of ASCs to differentiate into osteoblasts [46]. Of note, a link has previously been established between CD and a lack of cell differentiation, including osteoblast differentiation [47], suggesting that cell differentiation is worsened by smoking in people with CD.

In the second approach, DMRs were examined to know which regions of genes were more differentially methylated. Our analysis highlighted TNXB as having the most DMRs in ASCs from both smoking and ex-smoking patients as compared with ASCs from non-smoking patients. TNXB has been involved in collagen organization and matrix integrity [48]. Consistent with our results, Barrow and colleagues [49] reported that TNXB is highly differentially methylated in active smokers diagnosed with colorectal cancer. TNXB was found to be differentially methylated in smoker but also in ex-smoker ASCs from patients with CD compared to the non-smokers’ group. TNXB deficiency is associated with Classical-like Ehlers–Danlos syndrome (clEDS), which comprises a clinically and genetically heterogeneous group of connective tissue disorders [50]. Interestingly, Green and colleagues [51] proposed that patients with clEDS are predisposed to gastrointestinal tract tissue fragility. Consistent with this, smoking status was found to alter TNXB gene methylation pattern. As a result, it was thought that the TNXB gene may have an important role in maintaining gut homeostasis and restoring intestinal tissue damage, both of which processes are impaired in the context of CD.

Notably, some of the obtained processes in the DMR analysis were coincident with those of the DMPs, including biological processes implicated in the immunosuppressive properties of ASCs that are affected by CD such as wound healing, osteoblast differentiation, or response to transforming growth factor beta.

Our study has some limitations that warrant discussion. It will be necessary to validate our observational findings in a larger cohort and it would be important to use donor macrophages or lymphocytes instead of cancer-derived cell lines to reaffirm our results. Also, research that can examine patients with varying cigarette consumption per day or that can stratify patients depending on how long they have smoked (as well as patients who have quit smoking for various lengths of time) should be assessed to confirm our findings. The main strength of the present study is that it is the first, to our knowledge, to investigate the impact of smoking on the therapeutic properties of ASCs from patients with CD. Although challenging, it would be interesting to investigate specifically those tobacco metabolite(s) that might be responsible for these modifications in the immunological modulatory capabilities of ASCs. Finally, although the present study centered on SAT, future studies should also focus on the impact of smoking on visceral adipose tissue. Indeed, the mesenteric adipose tissue that surrounds the inflamed intestinal region, commonly known as creeping fat, actively participates in the generation and maintenance of the inflammatory microenvironment near the inflamed intestinal area [52]. For this reason, we believe smoking may also have a detrimental effect on the creeping fat and ASCs within this tissue. This would help to explain, at least in part, why smoking has a negative impact on CD prognosis and increases the risk of post-operative recurrence.

In sum, the present study provides evidence that smoking habit (current and former) in patients with CD is critical for the immune-modulating properties of ASCs from the SAT. We propose that this could be explained by epigenetic changes that remain imprinted in the DNA of ASCs even when the patient has quit smoking. Thus, we believe smoking and ex-smoking patients with CD should not be recommended for autologous ASC transplantation and, instead, allogenic transplantation from healthy individuals (who have never smoked) should be considered. To optimize cell therapies, however, the effects of smoking on the immunoregulatory properties of healthy ASCs deserve further study.

## 5. Conclusions

Cigarette smoking disturbs the immune regulatory properties of ASCs from the SAT of patients with CD and provokes an inflammatory phenotype. Smoking induces epigenetic modifications related to the therapeutic properties in these cells, and these modifications remain in ex-smokers.

## Figures and Tables

**Figure 1 cells-12-01021-f001:**
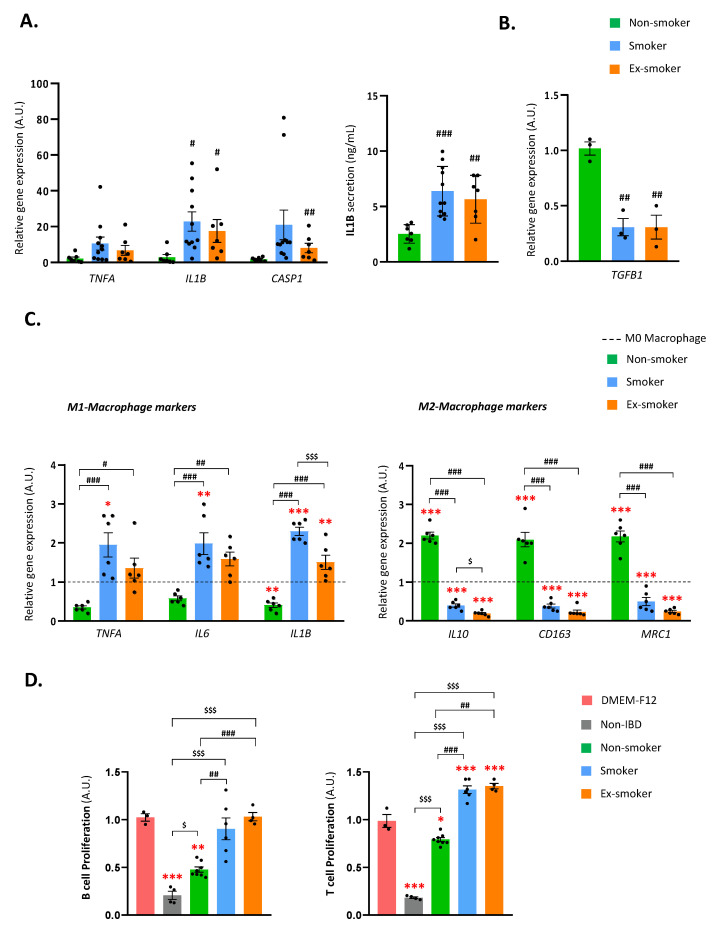
Adipose stem cells isolated from smokers and ex-smokers with Crohn’s disease have altered immunosuppressive properties. (**A**) Gene expression of pro-inflammatory genes (*TNFA*, *IL1B* and *CASP1*) (left panel) and secretion levels of the pro-inflammatory cytokine IL1B (right panel) in SAT-derived adipose stem cells (ASCs) obtained from non-smokers, smokers or ex-smokers with Crohn’s disease. # *p* < 0.05; ## *p* < 0.01 vs. non-smoker group. (**B**) Gene expression of *TGFB1* in ASCs obtained from non-smokers, smokers or ex-smokers with Crohn’s disease. ## *p* < 0.01; ### *p* < 0.001 vs. non-smoker group. (**C**) Gene expression of M1 macrophage markers (*TNFA*, *IL6* and *IL1B*) (left panel) and M2 macrophage markers (*IL10*, *CD163* and *MRC1*) (right panel) in THP-1 monocytes co-cultured or not (basal condition) with conditioned medium of ASCs obtained from non-smoker, smoker or ex-smoker patients with Crohn’s disease. * *p* < 0.05; ** *p* < 0.01; *** *p*< 0.001 vs. M0 macrophage (control). # *p* < 0.05; ## *p* < 0.01; ### *p* < 0.001 vs. non-smoker group. $ *p* < 0.05; $$$ *p* < 0.001 vs. smoker group. (**D**) Proliferation of B-cells (MEC-1 cell line) and T-cells (JURKAT cell line) co-cultured with conditioned medium of ASCs obtained from non-IBD subjects and from non-smoker, smoker and ex-smoker Crohn’s disease patients. * *p* < 0.05; ** *p* < 0.01; *** *p*< 0.001 vs. DMEM-F12 (control). ## *p* < 0.01; ### *p* < 0.001 vs. non-smoker group. $ *p* < 0.05; $$$ *p* < 0.001 vs. non-IBD group. Results are shown as mean ± SEM from independent donors’ experiments. Comparisons among the three groups were performed using two-way ANOVA followed with the Kruskal–Wallis test.

**Figure 2 cells-12-01021-f002:**
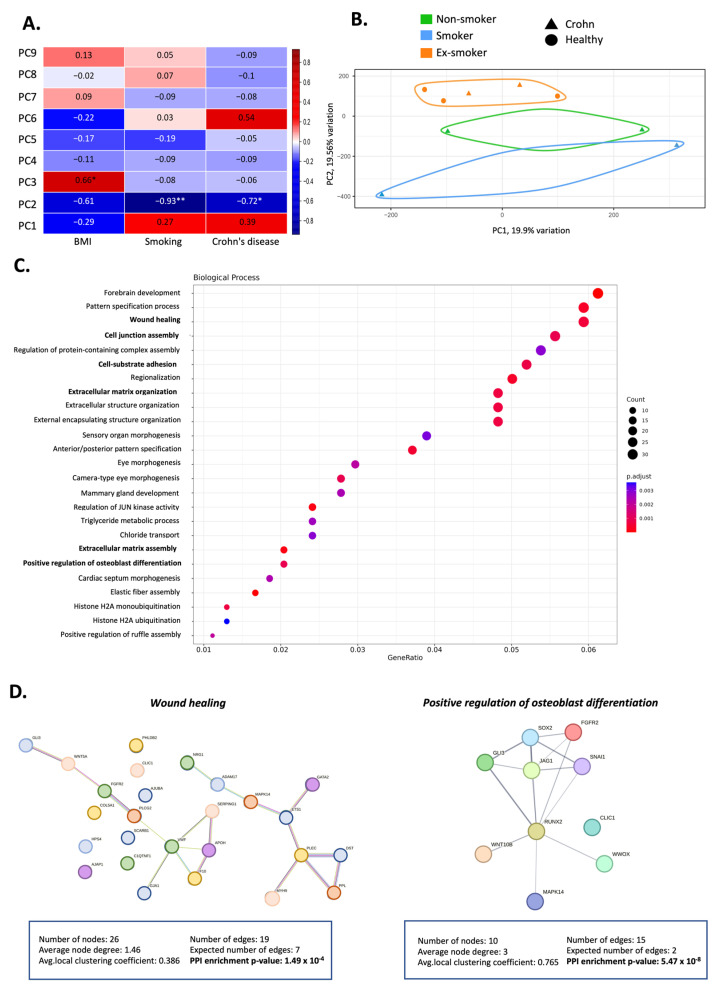
Exploratory data analysis of the differentially methylated positions in adipose-derived stem cells isolated from patients with Crohn’s disease and healthy controls. (**A**) Heat map showing the correlation data between the nine main components and clinical parameters such as body mass index (BMI), Crohn’s disease or smoking habit. Each cell of the heatmap shows the Pearson correlation coefficient. Asterisks indicate significant correlation (* *p* < 0.05; ** *p* < 0.01). (**B**) Principal component (PC) analysis demonstrating the first two principal components of the overall DNA methylation profiles, showing a discernible differentiation between patients with Crohn’s disease who are smokers (blue triangles), non-smokers (green triangles) or ex-smokers (orange triangles). Also, healthy controls are represented with a circle. (**C**) Gene ontology enrichment analysis of the biological processes using the whole set of genes with significant differentially methylated positions (DMPs). Different color scale represents the *p*-adjusted values and the size of the circles represents the number of genes implicated in each process. (**D**) Functional analysis of gene-associated DMPs in Crohn’s disease. Left. Gene–gene interaction involving 26 genes implicated in wound healing and which are differentially methylated in smoker, non-smoker and ex-smoker patients with Crohn’s disease. Right. Gene–gene interaction involving 10 genes implicated in positive regulation of osteoblast differentiation and which are differentially methylated in smoker, non-smoker and ex-smoker patients with Crohn’s disease.

**Figure 3 cells-12-01021-f003:**
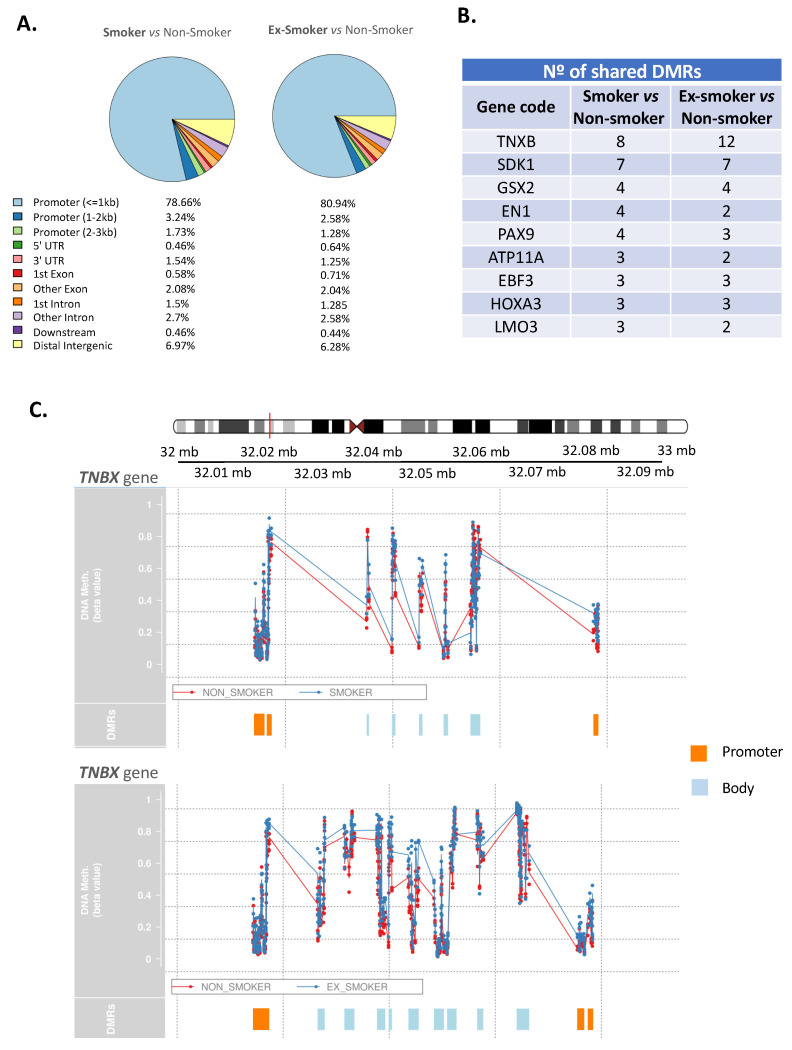
Smoking status revealed notable changes in differentially methylated regions in adipose-derived stem cells from patients with Crohn’s disease. (**A**) Intragenic differentially methylated region (DMR) distribution: 83.63% (smokers versus non-smokers) and 84.8% (ex-smokers versus non-smokers) of the DMRs are located in the promoter region and 15% are in the gene body. (**B**) Table representing genes with more DMRs which are present in both smoker vs. non-smoker and ex-smoker vs. non-smoker groups. (**C**) Significant DMRs of TNBX gene in the comparison smoker vs. non-smoker (panel above, 8 significant DMRs) and ex-smoker vs. non-smoker (panel below, 12 significant DMRs).

**Figure 4 cells-12-01021-f004:**
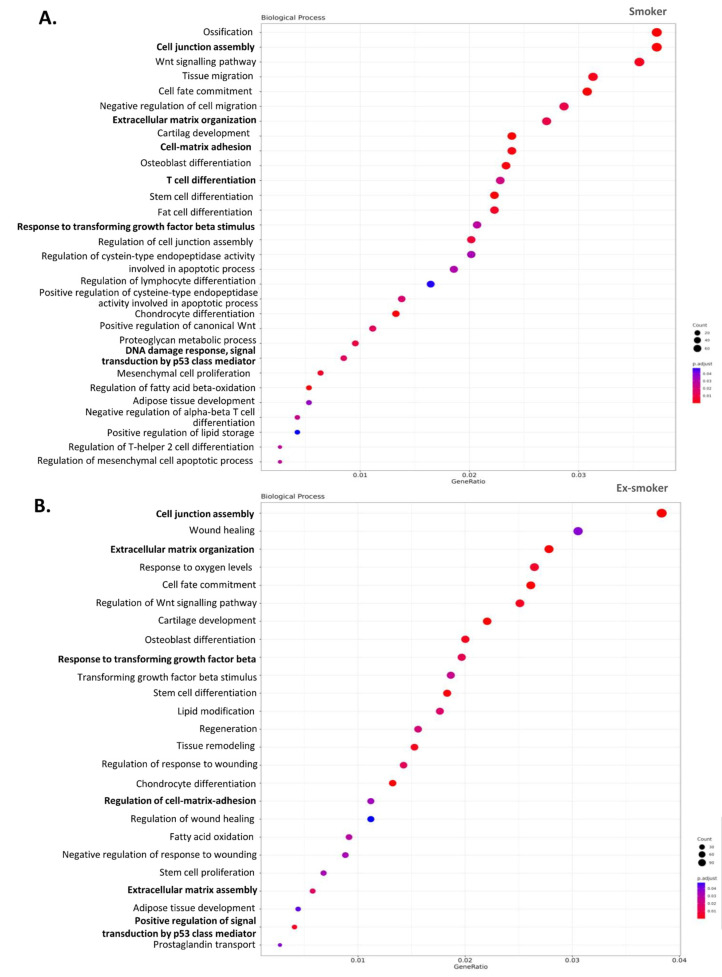
Cigarette smoking (current and past) disrupts the therapeutic properties in adipose stem cells from patients with Crohn’s disease. (**A**) Gene ontology (GO) enrichment analysis of the biological processes using the whole set of genes with significant DMRs obtained from the smoking group of patients and bubble plot for selected GOs. Different color scale represents the *p*-adjusted values and the size of the circles represents the number of genes implicated in each process. (**B**) GO enrichment analysis of the biological processes using the whole set of genes with significant DMRs obtained from the ex-smoking group of patients and bubble plot for selected GOs.

## Data Availability

The accession number for DNA methylation data reported in this paper is GSE138311.The other datasets generated during and/or analyzed during the current study are available from the corresponding author on reasonable request.

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
