# Peer review of "Smoking Suppresses the Therapeutic Potential of Adipose Stem Cells in Crohn’s Disease Patients through Epigenetic Changes"

_cells, 2023, doi:10.3390/cells12071021_

Round 1

Reviewer 1 Report

Cells

Cigarette smoking suppresses the therapeutic potential of adipose-stem cells of patients with Crohn’s disease through epigenetic modifications

Boronat-Toscano et al

This is an interesting look at differences in adipose stem cells from patients with CD, stratified by smoking status. The suggestions are that smoking history can impact the effectiveness of ASCs in terms of inflammatory outcomes, and that this basis may be in the epigenetic changes in ASCs driven by smoking. There are numerous events that suggest this, as found in this study, but very little mechanistic connection. Also, the sample sizes are quite small, and the authors note this as a study limitation. The findings are nonetheless interesting and suggestive. My comments and questions are found below.

Intro

Lines 70-71: It looks like something was missing at the end of the sentence

“For instance, obesity is 70 associated with mesenchymal stromal cell senescence [12] whereas physical exercise [13].”

Materials and Methods

Line 98: Supp Table S1 – does “Active/Inactive” refer to disease status? Also, define PCR (also in Supp Table 4) since you use polymerase chain reaction as a technique.

Line 109: T75 *flask*

Section 2.3: neither here nor in the references does it indicate how cells were acquired off plates for immunophenotyping (ie, were the cells trypsin-treated, which might remove target surface antigens?). Also, what are the sources of the antibodies, catalog numbers, dilutions, incubation times/conditions, etc?

Sections 2.5-2.6: Were the RNA extractions from THP-1 cells the same as those for the ASCs?

Line 199: is this missing +/- ahead of SEM?

Results

Section 3.1, Fig 1B, were there any positive control proliferation stimulators for the various immune cells? It would be curious to see how this compares to known amounts of proliferation from conventional stimulation such as phorbol esters for the Jurkat cells, etc.

Section 3.2, Figure 2B: this PCA depiction needs a diagnostic scree plot for a more constructive analysis.

Fig 2C also includes cell-cell and cell-matrix (and extracellular matrix [ECM]) interactions that could constitute wound healing processes.

For Supp Excel File 1 (and 2 and 3), could you write out “Biologic Process”; “Molecular Function”; and “Cellular Component” on the file tabs, as that would be easier to grasp.

Section 3.4, Supp XL files 2 and 3, for the Biologic Processes and Cellular Component GO analyses, there are again strong implications of cell-cell, cell-matrix and ECM involvement that could have implications in the processes noted in lines 319-321 as well as all of the Biologic Processes found in Fig 4.

 The authors are essentially using the ASCs from non-smokers with CD as controls. Do the authors have any data to suggest if ASCs from healthy donors differ dramatically (in whatever assays they have used) from the patient-derived ASCs? One sees how autologous ASC transplant would likely be the "simplest" mode in a therapeutic approach, but their data suggest that smoking status/history make nullify this. Consequently, does there appear to be a benefit in using healthy donor (albeit, allogeneic) ASCs based on an improved phenotype?

In Discussion/Conclusion, it must be mentioned that the recipient cells of the ASC-conditioned media are all cancer-derived cell lines whose properties may not completely reflect those of their non-cancerous counterparts. It could be suggested that further studies beyond this manuscript should utilize donor macrophage, T- and B-cell types (and their various subsets). This is particularly relevant in the potential generation of regulatory T cells in the context of TGFB, given the authors’ results suggesting that this may be deregulated according to GO studies (Fig 4). One also notes the “T cell differentiation” and “Regulation of T-helper 2 cell differentiation” GO categories there. This fits with the statement in Line 387: “Of note, a link has previously been established between CD and a lack of cell differentiation, including osteoblast differentiation [42], suggesting that cell differentiation is worsened by smoking in people with CD.”

Do the authors have any transcriptomic data for TGFB family members from the different ASC donor types (re: smoking status) or direct ELISA measurements of TGFB (1, 2, and/or 3) in the ASC conditioned media? This would make a nice tie from the epigenetic data to the immune cell data in Fig 1. Of course, TGFB also plays roles in extracellular matrix modification as noted in some comments above where those pathways seem to be prevalent in the Biologic Processes. You also note that ETS1 is involved in TGFB pathway regulation (lines 378-379). With all of these potential mechanistic connections to TGFB, it seems this should be explored a bit further, as mentioned above. One anticipates it would be highest in non-smoker ASCs (either mRNA, protein forms, or released extracellularly), less in ex-smoker ASC, and relatively lowest in ASCs from smokers.

Are there any suggestion of enhanced (or decreased) epigenetic modifiers (methyl transferases/methylases) in CD non-smokers vs smokers vs ex-smokers?

As the authors demonstrate, ASC secretomes may mediate the stem cell effects. Extracellular vesicles are likely prominent components of those secretomes and may be the functional mediators of stem cell effects (PMID: 30852701; PMID: 32642003). Do they authors have any insights on the potential differences in ASC EVs from different sources with respect to smoking status, particularly in the context of CD? The EVs may have more relevance than the stem cells themselves in terms of therapeutic potential, particularly in allogeneic settings.

Reviewer 2 Report

Dear Editor, 

The title is interesting and can be consider for publication. Authors investigated the smoking induces epigenetic changes that impair the capacity of adipose-stem cells (ASCs) to suppress the immune system. There are some comments for improving the current manuscript:

-Title should be shorten

-It would be better to rewrite the sentences that contain "we"

-Introduction needs to be improved in epigenetic description and approaches, it would be use the following reference;

https://doi.org/10.1155/2022/5304860 (https://www.hindawi.com/journals/sci/2022/5304860/)

-Authors should express the aim of study at end of introduction

-Method and materials are ok, but authors need to explain the details of the below subtitles:

2.3. Adipose stem cell immunophenotyping

2.6. RNA extraction

2.7. Real-time quantitative PCR 

-Results are well-organized and explained

-It would be better to add new studies (2015-2023) in discussion section

Best Regards,  

Reviewer 3 Report

The manuscript by Boronat-Toscano et al., examines the effect of cigarette smoking on the immunosuppressive capabilities of Adipose Stem Cells (ASC). Overall, the conclusions are supported by the data. 

Minor recommendations-

Line 71- ends abruptly. 

Line 214 and figure 1- statistical analysis should be explained more clearly. For e.g., is there a significant difference between CASP1 gene expression between smokers and non-smokers? 

Author Response

Response to Referee 3

The manuscript by Boronat-Toscano et al., examines the effect of cigarette smoking on the immunosuppressive capabilities of Adipose Stem Cells (ASC). Overall, the conclusions are supported by the data.

Minor recommendations-

Line 71- ends abruptly.

Sorry for this mistake. We have finished the sentence (Page 2, lines 71-72).

Line 214 and figure 1- statistical analysis should be explained more clearly. For e.g., is there a significant difference between CASP1 gene expression between smokers and non-smokers?

We really appreciate your suggestion. We have included the statistical analysis in the legend of Figure 1. We did not appreciate significant differences between smokers and non-smokers group (p value 0.0864).

Round 2

Reviewer 1 Report

The authors have responded well to my critiques.

Reviewer 2 Report

Dear Editor,

The revised manuscript is acceptable for publication. 

Best Regards,